

# Cytogenetic analyses in *Trinomys* (Echimyidae, Rodentia), with description of new karyotypes

Naiara Pereira Araújo[1], Cayo Augusto Rocha Dias[1,2], Rodolfo Stumpp[1] and Marta Svartman[1]

[1] Departamento de Biologia Geral, Universidade Federal de Minas Gerais, Belo Horizonte, Minas Gerais, Brazil
[2] Departamento de Zoologia, Instituto de Ciências Biológicas, Universidade Federal de Minas, Belo Horizonte, Minas Gerais, Brazil

## ABSTRACT

*Trinomys* Thomas (1921) is a terrestrial genus of spiny rats endemic to the Brazilian areas of Atlantic Forest and the transitional areas of Cerrado and Caatinga. Although most species have been already karyotyped, the available cytogenetic information is mostly restricted to diploid and fundamental numbers. We analyzed the chromosomes of two *Trinomys* species: *Trinomys moojeni* ($2n = 56$, FN = 106) and *Trinomys setosus setosus* ($2n = 56$, FN = 106 and $2n = 56$, FN = 108). Our analyses included GTG- and CBG-banding, silver-staining of the nucleolar organizer regions, and chromosome mapping of telomeres and 45S rDNA by fluorescent *in situ* hybridization (FISH). Comparative GTG- and CBG-banding suggested that the interspecific variation may be due to rearrangements such as pericentric inversions, centromere repositioning, and heterochromatin variation. We report two new karyotypes for *T. s. setosus* and describe for the first time the banding patterns of the two *Trinomys* species.

## INTRODUCTION

Spiny rats (family Echimyidae) are the most diverse group of South American hystricognath rodents. There are 22 extant genera and around 90 species found from Central America to Northern Argentina, where they have radiated across multiple biomes, including a vast array of ecomorphological adaptations, encompassing arboreal, semi-fossorial, terrestrial, and semi-aquatic lifestyles (*Emmons, Leite & Patton, 2015*). The great variation in their life history, adaptations, and morphotypes also extends to their karyotypes. Their diploid numbers ($2n$) range from $2n = 14$ in *Proechimys* gr. *longicaudatus* (*Amaral et al., 2013*) to $2n = 118$ in the arboreal species *Dactylomys boliviensis* (*Dunnum, Salazar-Bravo & Yates, 2001*), which has the highest $2n$ known among mammals. This variation results from the presence of B-chromosomes (*Yonenaga-Yassuda et al., 1985*; *Fagundes, Camacho & Yonenaga-Yassuda, 2004*), multiple sex chromosome systems (*Amaral et al., 2013*; *Costa et al., 2016*), and several rearrangements, including inversions, fusions/fissions, and constitutive heterochromatin variation. The great karyotypic variability observed in Echimyidae represents an opportunity to elucidate mechanisms of chromosome evolution and their role during speciation and diversification.

Corresponding authors
Naiara Pereira Araújo, naiaraparaujo@yahoo.com.br
Marta Svartman, svartmanm@ufmg.br

**Table 1  Karyotypic data of species of *Trinomys*.**

| Species[a] | 2n/FN | Banding patterns/FISH | References |
|---|---|---|---|
| *Trinomys albispinus* | | | |
|    *T. a. albispinus* | 60/116 | Ag-RONs | *Souza, Corrêa & Pessôa (2006)* |
|    *T. a. minor* | 60/116 | GTG, CBG, RBG, Ag-NORs | *Leal-Mesquita et al. (1992)* |
| *Trinomys dimidiatus* | 60/116 | – | *Pessôa et al. (2005)* |
| *Trinomys eliasi* | 58/112 | – | *Pessôa et al. (2005)* |
| *Trinomys gratiosus* | | | |
|    *T. g. gratiosus* | – | – | – |
|    *T. g. bonafidei* | 56/108 | – | *Pessôa et al. (2005)* |
|    *T. g. panema* | – | – | – |
| *Trinomys iheringi* | 61–65/116 | GTG, CBG, RBG, Ag-NORs, FISH with telomeric and rDNA probes | *Yonenaga-Yassuda et al. (1985)* *Fagundes, Camacho & Yonenaga-Yassuda (2004)* |
| *Trinomys mirapitanga* | – | – | – |
| *Trinomys moojeni* | 56/106 | – | *Corrêa et al. (2005)* |
| *Trinomys paratus* | 58/112 | CBG | *Lazar et al. (2017)* |
| *Trinomys setosus* | | | |
|    *T. s. elegans* | 56/104 | – | *Corrêa et al. (2005)* |
|    *T. s. setosus* | – | – | – |
| *Trinomys yonenagae* | 54/104 | GTG, CBG, RBG, Ag-NORs | *Leal-Mesquita et al. (1992)* |

**Notes.**
[a]Classification based on *Pessôa et al. (2015)*.
2n, diploid number; FN, fundamental number.

Within Echimyidae, the Atlantic spiny rats of the genus *Trinomys* Thomas, 1921, allocated within Euryzygomatomyinae (*Lara & Patton, 2000*; *Fabre et al., 2017*), are amongst the most taxonomically complex genera. *Trinomys* comprises ten extant species endemic to Brazilian areas of Atlantic Forest and transitional areas of Cerrado and Caatinga (*Pessôa et al., 2015*). Most species have few morphological synapomorphies, with many primitive and few derived features (*Dalapicolla & Leite, 2015*), which led different authors to consider several of them as subspecies in different taxonomic arrangements (*Lara, Patton & Da Silva, 1996*; *Lara & Patton, 2000*; *Pessôa et al., 2015*). Three species, *Trinomys eliasi*, *Trinomys moojeni*, and *Trinomys yonenagae*, are considered near threatened or endangered due to forest fragmentation and habitat destruction (http://www.iucnredlist.org).

As for most rodents, *Trinomys* presents a confusing taxonomic history. Until 1996, it was considered a subgenus of *Proechimys* due to craniodental and body similarities (*Moojen, 1948*; *Lara, Patton & Da Silva, 1996*). *Trinomys* was then raised to a generic level after studies including biogeographic data, dental characters, and mitochondrial DNA sequence-based phylogenies (*Lara, Patton & Da Silva, 1996*; *Lara & Patton, 2000*; *Carvalho & Salles, 2004*). More recently molecular phylogenetic studies with mitochondrial and nuclear sequences strongly supported *Trinomys* as a sister taxon to *Clyomys* and *Euryzygomatomys*, excluding its relationship with *Proechimys* (*Fabre et al., 2012*; *Fabre et al., 2017*; *Upham & Patterson, 2012*).

The karyotypes of all recognized species of *Trinomys* have already been described, with the exception of *Trinomys mirapitanga* (Table 1). Nevertheless, most reported cytogenetic

**Table 2** Specimens analyzed.

| Species | 2n | FN | Collection sites | Deposit numbers (sex) | GenBank accession numbers |
|---|---|---|---|---|---|
| *Trinomys moojeni* | 56 | 106 | Serra do Caraça Private Reserve/MG (20°05′S, 43°29′W) | MCN-M 2816 (F) | KX650080.1 |
| *Trinomys setosus setosus* | 56 | 106 | Serro/MG (18°36′S, 43°22′W) | UFMG 6024 (F) | KX655539.1 |
| | 56 | 108 | Morro do Pilar/MG (19°15′S, 43°19′W) São Gonçalo do Rio Abaixo/MG (19°49′S, 43°21′W) | MCN-M 3296 (F)/ MCN-M 3297 (F) MCN-M 2587 (M) | MG214347/ MG214348 MG214349 |

**Notes.**
2n, diploid number; FN, fundamental number; M, male; F, female; MCN-M, Museu de Ciências Naturais—Pontifícia Universidade Católica (PUC, Minas Gerais, Brazil); UFMG, Centro de Coleções Taxonômicas—Universidade Federal de Minas Gerais (CCT-UFMG, Minas Gerais, Brazil).

data are restricted to the description of the 2n and fundamental numbers (FN), without information on banding patterns or FISH. The 2n ranges from 2n = 54 in *T. yonenagae* to 2n = 60 in *Trinomys albispinus, Trinomys dimidiatus,* and *Trinomys iheringi* (Table 1). Some specimens of *T. iheringi* presented a higher 2n due to the presence of minute supernumerary chromosomes (*Yonenaga-Yassuda et al., 1985*; *Fagundes, Camacho & Yonenaga-Yassuda, 2004*). Comparisons of the GTG-banded chromosomes of *T. iheringi, T. albispinus minor* (2n = 60, FN = 116), and *T. yonenagae* (2n = 54, FN = 104), the only *Trinomys* species analyzed after banding, revealed very conserved karyotypes (*Leal-Mesquita et al., 1992*). Closely related species of *Trinomys* seem to share the same karyotype, as is the case of the sister taxa *Trinomys paratus* and *T. eliasi* (both with 2n = 58, FN = 112), and of *T. dimidiatus* and *T. iheringi* (both with 2n = 60, FN = 116). In fact, it has been suggested that the divergence time among *Trinomys* species was not sufficient to produce great karyotypic changes (*Souza, Corrêa & Pessôa, 2006*; *Lazar et al., 2017*).

We comparatively analyzed the karyotypes of *T. moojeni* and *T. s. setosus*, including GTG- and CBG-banding, silver staining of the nucleolar organizer regions (Ag-NORs), and FISH with telomeric and 45S rDNA probes. Two new karyotypes are described for *T. s. setosus* and this is the first description of banding patterns for both species.

## MATERIAL AND METHODS

We studied five specimens of *Trinomys*, collected in the state of Minas Gerais, southeastern Brazil (Table 2), under the permits provided by the Instituto Chico Mendes de Conservação da Biodiversidade (ICMBio; permit number 22279-1 to Beatriz Dias Amaro) and the Instituto Brasileiro do Meio Ambiente e dos Recursos Naturais Renováveis (SISBIO-IBAMA; permit numbers 12989-2 and 36574-1, concedido a Adriano Pereira Paglia and Fabíola Keesen Ferreira, respectively). The conducted research was approved by the Ethics Committee in Animal Experimentation (CEUA) of Universidade Federal de Minas Gerais (approval number: 211/2013). Voucher specimens were deposited in the Museu de Ciências Naturais da Pontifícia Universidade Católica de Minas Gerais (MCN-M, Minas Gerais, Brazil) or in the Centro de Coleções Taxonômicas da Universidade Federal de Minas Gerais (CCT-UFMG, Minas Gerais, Brazil). Morphological identification was based

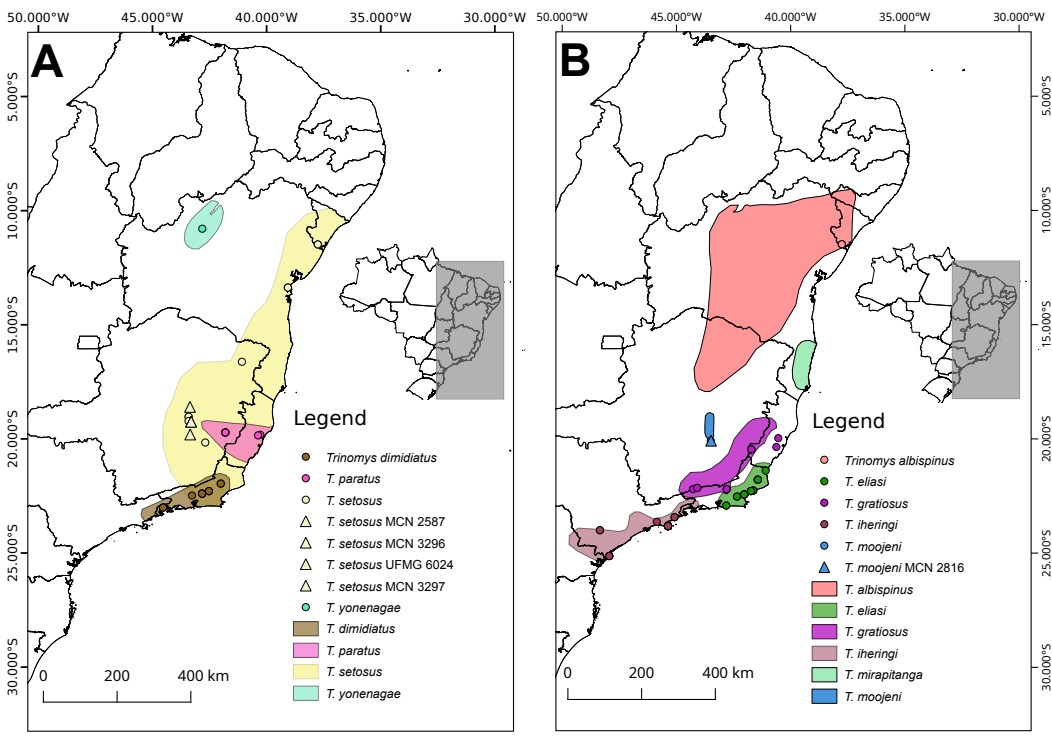

**Figure 1** **Sample sites of the specimens used in this study against the known range of *Trinomys* species.** For detailed references on spatial data downloaded from IUCN for each species, see Supplemental Information 1. Distribution of (A) *T. dimidiatus*, *T. paratus*, *T. setosus*, and *T, yonenagae*, and (B) *T. albispinus*, *T. eliasi*, *T. gratiosus*, *T. iheringi*, *T. mirapitanga*, and *T. moojeni*. Triangles represent our specimens and circles indicate the specimens whose mitochondrial sequences were retrieved from GenBank.

on skull, dental and skin characters described by *Moojen (1948)*, *Iack-Ximenes (2005)*, *Dalapicolla & Leite (2015)* and *Pessôa et al. (2015)*. The morphological diagnosis of each specimen is given in Supplemental Table S1. We plotted the sampling sites of the specimens used in this study against the known range of *Trinomys* species using QGIS 2.18.16 (Fig. 1; *QGIS Development Team, 2018*). Spatial datasets containing the known range of *Trinomys* species were obtained from IUCN (http://www.iucnredlist.org).

Chromosome preparations were obtained directly from bone marrow (*Ford & Hamerton, 1956*). GTG- and CBG-banding patterns and silver-staining of the nucleolar organizer regions (Ag-NORs) were performed according to *Seabright (1971)*, *Sumner (1972)*, and *Howel & Black (1980)*, respectively. FISH with a biotinylated telomeric sequence (Invitrogen, Carlsabad, CA, USA) and with the R2 45S rDNA probe labeled by nick translation with digoxigenin-11-dUTP (DIG-Nick Translation mix; Roche Applied Science, Penzberg, Germany), followed *Araújo et al. (2017)*; *Araújo et al. (2014)*, respectively. Immunodetection was carried out with neutravidin and antidigoxigenin, both conjugated with rhodamine (Roche Applied Science). The analyses and image acquisition were performed under a Zeiss Axioimager 2 epifluorescence microscope using the AxioVision software (Zeiss, Oberkochen, Germany), Adobe Photoshop CS3

Extended was used for image edition. For each specimen, at least 20 metaphases of each experiment were analyzed.

Ordination and phylogenetic methods were employed in order to check the assignment of MCN-M 2587, MCN-M 3296, and MCN-M 3297 to *T. setosus*. In order to do this, we sequenced the 401 bp-long segment of the mitochondrial cytochrome *b* (cyt*b*) of each specimen and included sequences from nine *Trinomys* species retrieved from GenBank in a phylogenetic analysis. The sequences of the specimens MCN-M2816 (*T. moojeni*) and UFMG 6024 (*T. s. setosus*) were previously deposited in GenBank after assembly of their mitochondrial genomes (*Araújo et al., 2016*). *Euryzygomatomys spinosus,* Fischer, 1814, was used as outgroup. Total genomic DNA of each *Trinomys* specimen was extracted from liver and their cyt*b* was amplified by polymerase chain reaction (PCR) with primers MVZ 05 and MVZ 04 (*Smith & Patton, 1993*). The PCR products were purified using the Wizard SV Gel and PCR Clean-up System kit (Promega, Madison, WI, USA) and sequenced on the ABI3130 platform (Myleus Biotechnology). The GenBank accession numbers of the sequences generated in this study, as well as those included in the analyses are presented in the Supplemental Information 1.

The sequences obtained and those from GenBank were aligned using the Muscle (*Edgar, 2004*) algorithm. MEGA 7 (*Kumar, Stecher & Tamura, 2016*) was used to build a Kimura-2-parameter corrected distance matrix in which our ordination analysis was based. Ordination methods were used since they are useful tools to perform dimensionality reduction and to represent the distance between sequences in a coordinate (Cartesian) space where the distances are preserved (*Higgins, 1992*; *Ramette, 2007*; *Zhang et al., 2011*). Principal Coordinate Analysis (PCoA) was used to explore the similarity among our specimen's sequences and other *Trinomys* species. The analysis was conducted in R (*R Core Team, 2017*) using the "pcoa" function in package APE (*Paradis, Claude & Strimmer, 2004*) and Lingoes procedure for correcting for negative eigenvalues.

Phylogenetic relatedness was used as a way of determining the most probable identity of the subject sequences. Thus, two methods of phylogenetic reconstruction were employed: maximum likelihood (ML) and bayesian inference (BI), which were carried out in RaxML 8 (*Stamatakis, 2014*) and MrBayes 3.2 (*Ronquist et al., 2012*), respectively. ML search comprised optimizations over 100 randomized maximum parsimony starting trees using the rapid hill-climbing algorithm under the GTRGAMMA model. As a measure of branch support, information on frequencies of 1,000 replicates of non-parametric bootstrap were annotated on the best-scoring ML tree. Bayesian inference comprised two independent runs composed of four chains each. A reversible jump MCMC sampling was used in order to explore different substitution schemes. Parameters and trees were sampled every 1,000 generations along a total of 20 million generations. After discarding a quarter of samples as burn-in, parameters and trees were summarized and the following metrics were used to assess MCMC convergence: standard deviation of split frequencies, effective sample size and potential scale reduction factor for each parameter.

## RESULTS

Bayesian inference and ML trees recovered *Trinomys* as monophyletic and grouped MCN-M 2816 within *T. moojeni* and the specimens UFMG 6024, MCN-M 2587, MCN-M 3296, and MCN-M 3297 within *T. setosus* (Fig. 2; Figs. S1 and S2). This was further supported by the PCoA results, that showed samples of the same species clustering together on the graph (Fig. S3). Morphological characters analyses corroborated the phylogeny and allowed assigning the *T. setosus* specimens as *T. s. setosus* (Table S1).

The female *T. moojeni* had a complement with $2n = 56$ and FN = 106, similar to that described by *Corrêa et al. (2005)*, composed of 26 pairs of biarmed (pairs 1–26) and one pair of acrocentric (pair 27) autosomes, and submetacentric X chromosomes (Fig. 3). The autosomes of the female *T. s. setosus* ($2n = 56$, $FN = 106$) collected in Serro included 26 biarmed pairs decreasing in size from large to small (pairs 1–25 and 27) and a small acrocentric pair (pair 26). The X chromosomes were large acrocentrics (Fig. 4A). The other three specimens of *T. s. setosus* had karyotypes with $2n = 56$ and FN = 108 (Fig. 4C and Fig. S4), similar to the other cytotype of *T. s. setosus*, but with pair 26 as a biarmed element. Their X chromosome was a large acrocentric and the Y was a small acrocentric (Fig. 4 and Fig. S4).

After GTG-banding, it was possible to identify all chromosomes of each species (Figs. 3A and 4) and to verify that the complements of *T. s. setosus* with $FN = 106$ and $FN = 108$ (Fig. 5) differed in relation to pair 26, which was acrocentric or biarmed, in the animals with $FN = 106$ and $FN = 108$, respectively. CBG-banding revealed weak centromeric constitutive heterochromatin in pairs 1–5, 9, 10, 15, 17–27, and the X chromosome of *T. moojeni* (Fig. 3B); pairs 1, 11, 15, 16, 19–27, and the X chromosome of *T. s. setosus* ($2n = 56$, $FN = 106$; Fig. 4B); and pairs 1, 6, 8, 16, 18–27, and the sex chromosomes of *T. s. setosus* ($2n = 56$, $FN = 108$; Fig. 4D). Both species had a large interstitial secondary constriction on the long arm of pair 6, which bears the NORs (Fig. 6). Hybridization with the telomeric probe showed signals only at the termini of all chromosomes of the two species studied (Figs. 6C, 6F and 6I).

## DISCUSSION

The interspecific grouping of *Trinomys*, recovered by the phylogenetic analyses, was congruent with previous studies (*Lara & Patton, 2000*; *Tavares, Pessôa & Seuánez, 2015*; *Lazar et al., 2017*). Our phylogenetic analysis is also supported by the specimens' distribution (Fig. 1), morphology (Supplemental Table S1) and karyotypes. The collecting locality of *T. moojeni* (MCN-M 2816), Serra do Caraça Private Reserve, is the same of the specimens studied by *Cordeiro-Júnior & Talamoni (2006)* and the karyotype was similar to that described for this species (*Corrêa et al., 2005*; Fig. 3). *T. s. setosus*, in turn, which occurs from the coastal area of the Brazilian states of Sergipe, Bahia, and Espírito Santo to the interior of Minas Gerais (*Pessôa et al., 2015*), were collected in three municipalities of Minas Gerais.

A comparison of the GTG-banded chromosomes of *T. moojeni* and *T. s. setosus* ($2n = 56$, FN = 106) evidenced very similar karyotypes. They mainly differed on their pairs 2, 27,

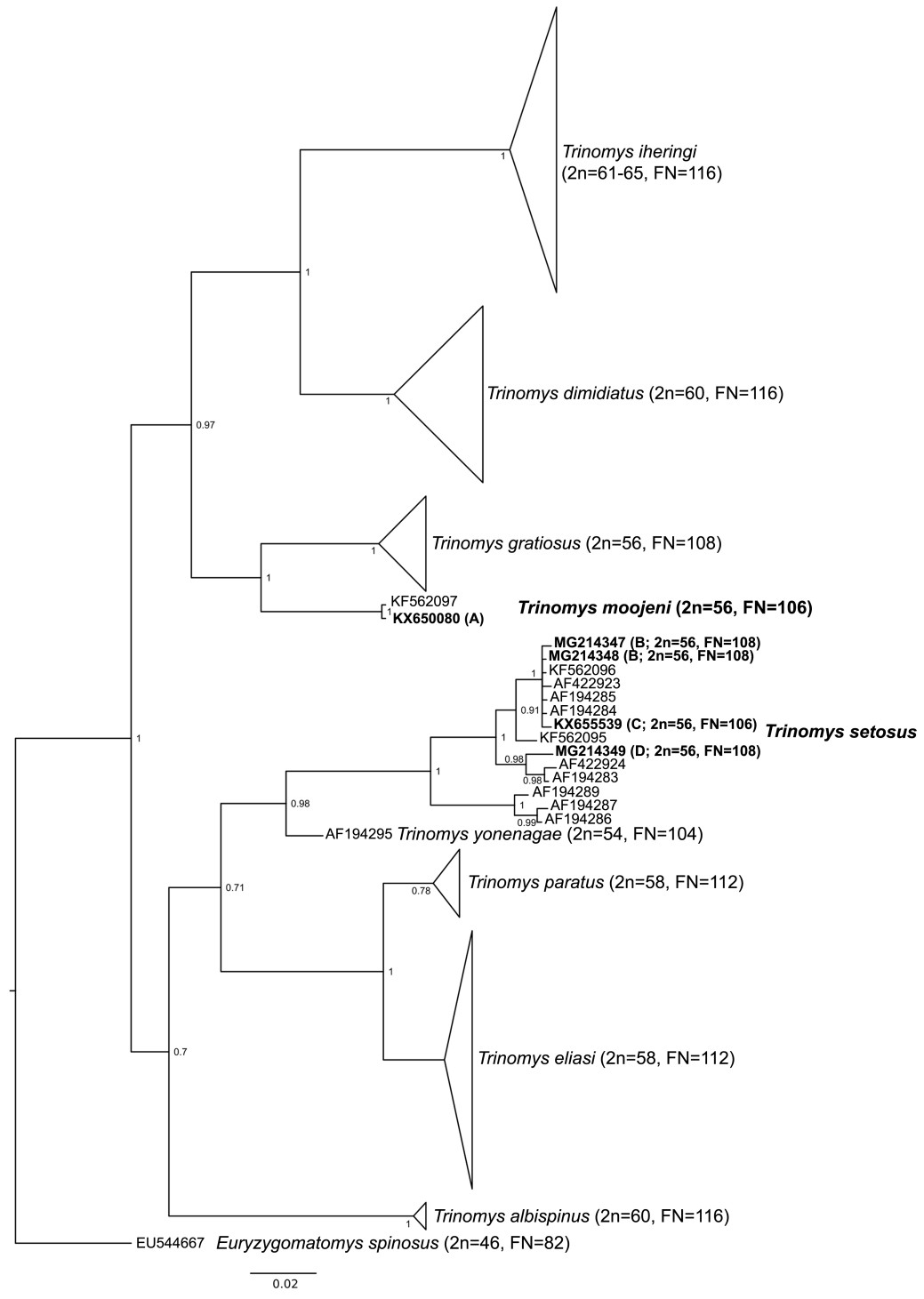

**Figure 2** **Collapsed bayesian inference tree based on a 401-bp fragment of the cytochrome *b* gene from species of *Trinomys*.** *E. spinosus* was used as outgroup. Collection sites: (A) Serra do Caraça Private Reserve/MG, (B) Morro do Pilar/MG, (C) Serro/MG, (D) São Gonçalo do Rio Abaixo/MG. Numbers represent Bayesian posterior probabilities ≥0.95. Specimens included in this study are in bold. See Fig. S1 for specimens details of the collapsed branches. Scale bar represents the number of substitutions per site.

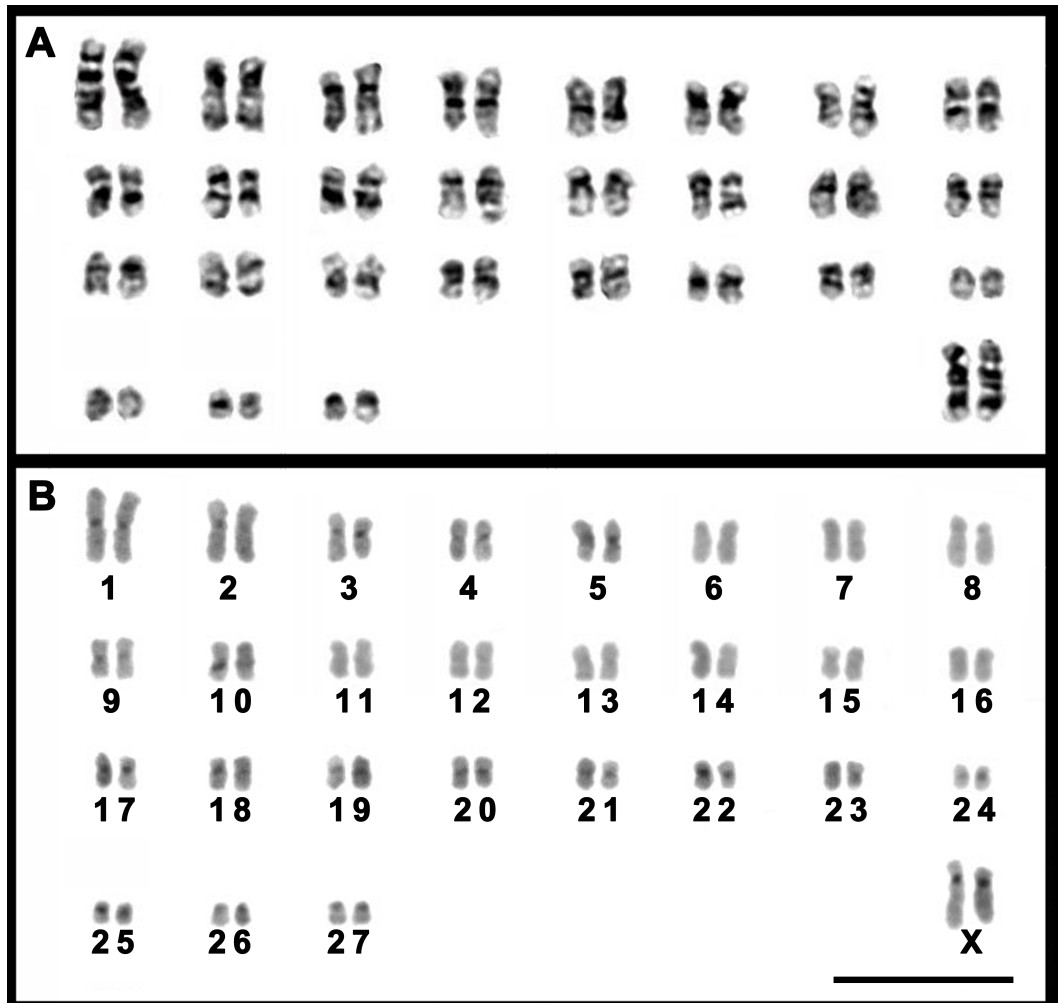

**Figure 3  Karyotypes of a female *T. moojeni* (2n = 56, FN = 106).** Karyotypes of a female *T. moojeni* (2n = 56, FN = 106) from Serra do Caraça Private Reserve, Minas Gerais State, after (A) GTG- and (B) CBG-banding. Scale bar = 10 μm.

and X chromosomes, possibly due to inversions and/or centromere repositioning (Fig. 5). In order to establish the exact mechanisms involved, further experiments including FISH with specific sequences from the regions of interest are necessary.

The karyotypes described herein for *T. s. setosus* differed in 2n, FN, and/or the sex chromosome morphology from those already published for this genus (*Yonenaga-Yassuda et al., 1985*; *Leal-Mesquita et al., 1992*; *Corrêa et al., 2005*; *Pessôa et al., 2005*; *Souza, Corrêa & Pessôa, 2006*; *Lazar et al., 2017*; Table 1). *Trinomys gratiosus bonafidei* also has 2n = 56 and FN = 108, but differently from our specimens, has a metacentric Y chromosome (*Pessôa et al., 2005*). The most recent revision on *Trinomys* divided *T. setosus* into the subspecies *T. s. setosus* and *T. s. elegans* (*Pessôa et al., 2015*). The diploid number was reported only for *T. s. elegans* and without banding patterns (2n = 56, FN = 104; *Corrêa et al., 2005*). *Pessôa et al. (2015)* mentioned that the karyotype of *T. s. setosus* from Almenara, Minas Gerais state,

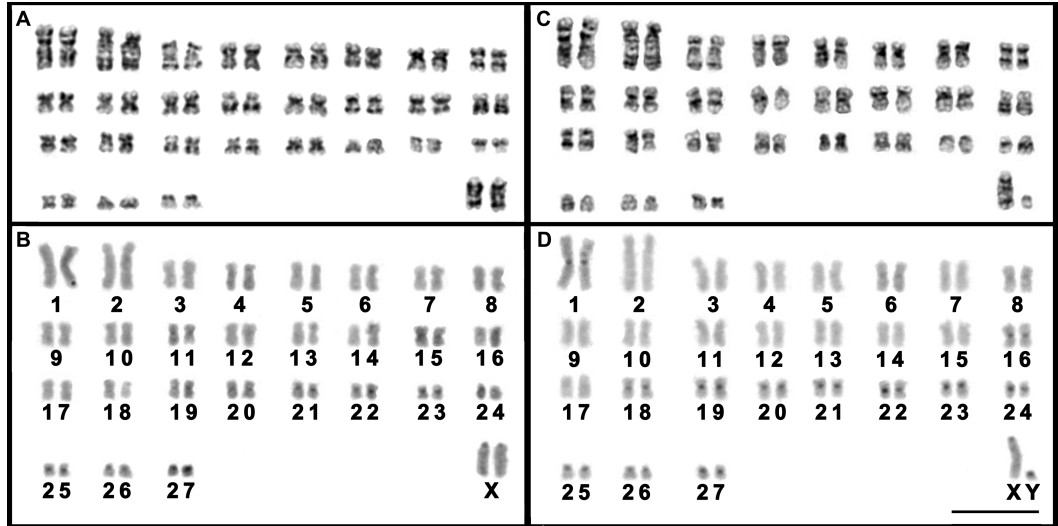

**Figure 4  Karyotypes of *T. s. setosus*.** A female with $2n = 56$, $FN = 106$ (A and B) from Serro, Minas Gerais state, and of a male with $2n = 56$, $FN = 108$ (C and D) from São Gonçalo do Rio Abaixo, Minas Gerais state, after GTG- (A and C) and CBG-banding (B and D). Scale bar = 10 μm.

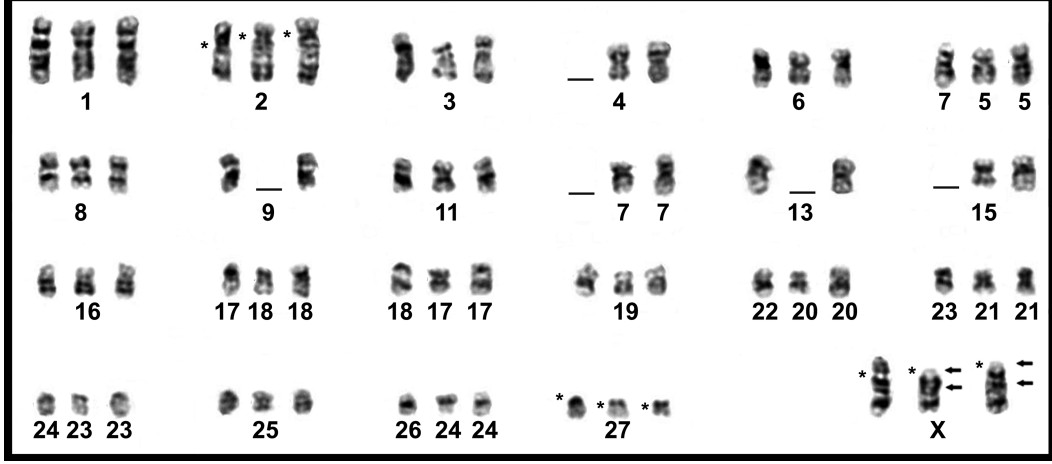

**Figure 5  Comparison of GTG-banded chromosomes of *Trinomys* species** Chromosomes from the left to the right: *T. moojeni* ($2n = 56$, $FN = 106$), *T. s. setosus* ($2n = 56$, $FN = 106$), and *T. s. setosus* ($2n = 56$, $FN = 108$). * = centromere position. The arrows indicate possible inversion sites.

has $2n = 56$ and FN = 108, but no figure was provided. Our *T. s. setosus* had karyotypes with FN = 106 and 108 and differed from that described by *Corrêa et al. (2005)* by the presence of additional short arms on pair 27 and pairs 26 and 27 of our specimens, respectively. These differences may be real or may reflect variations in chromosome condensation between both samples, as poorly elongated small chromosomes could prevent the detection of short arms. If real, these differences between *T. s. setosus* and *T. s. elegans* may be correlated with their subspecies allocation or may be due to interpopulational variation, as seems
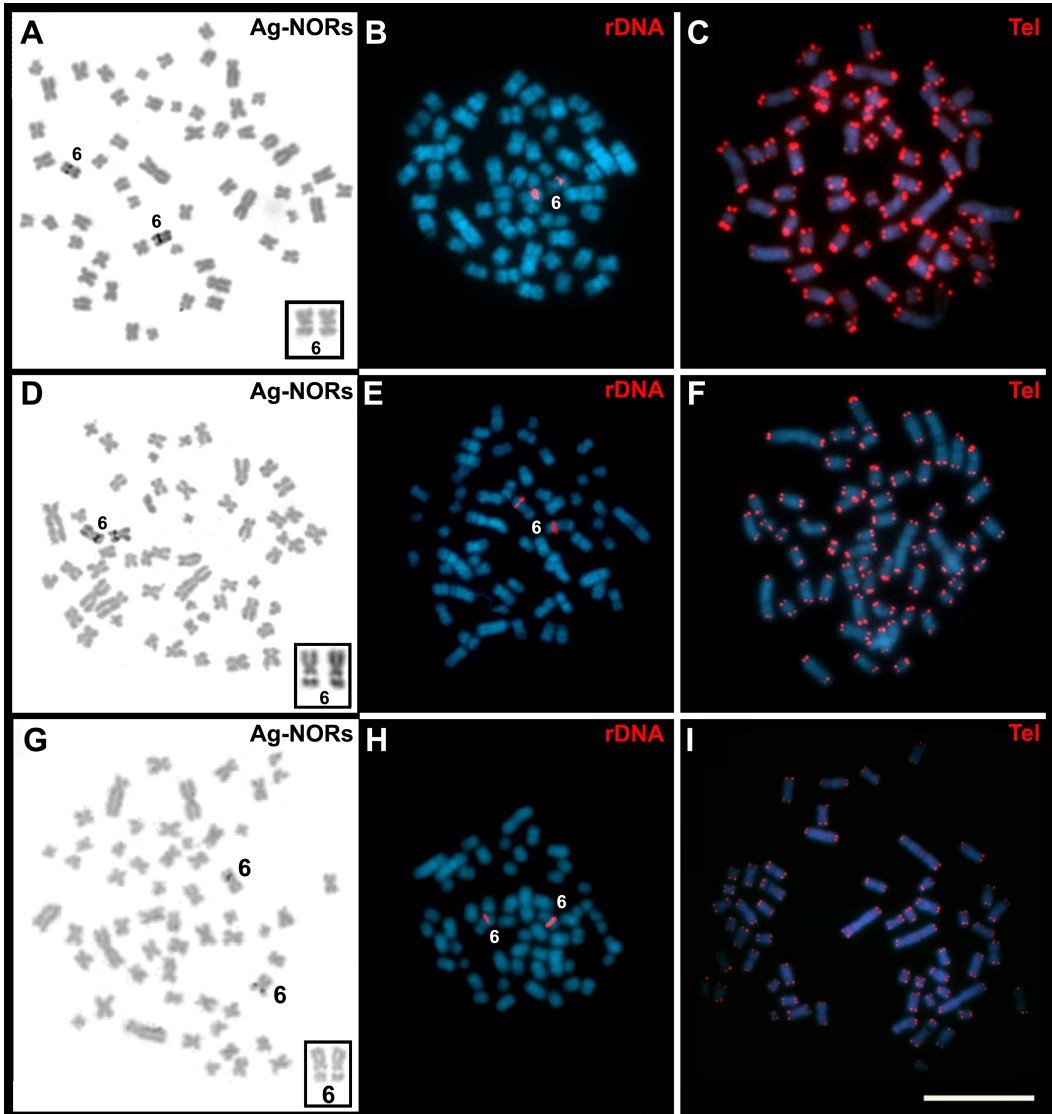

**Figure 6** **Cells of *Trinomys* species after Ag-NOR, FISH with the 45S rDNA, and telomeric probe (Tel).** Metaphases of (A–C) a female *T. moojeni* (2*n* = 56, *FN* = 106), (D–F) a female *T. s. setosus* (2*n* = 56, *FN* = 106), and (G–I) a male *T. s. setosus* (2*n* = 56, *FN* = 108). In the insets (A, D, and G), pair 6 after Giemsa staining. Note the secondary constrictions. Scale bar = 10 μm.

to be the case of *T. s. setosus*. Our phylogenetic analyses did not distinguish between *T. s. setosus* and *T. s. elegans* (Fig. 2, Figs. S1 and S2), but the morphological analysis allowed to recognize these subspecific taxa. The karyotype information was also relevant in species identification, revealing karyotypes that differed from those of other species of the genus.

The *T. s. setosus* karyotypes described herein also differed from others previously reported for *Trinomys* in the morphology of the X chromosome. With the exception of *T. setosus* and *T. yonenagae*, which presented acrocentric X chromosomes, all *Trinomys*

species had a submetacentric X (*T. albispinus, T. dimidiatus, T. eliasi, T. gratiosus, T. iheringi, T. moojeni,* and *T. paratus*). Based on our cytochrome *b* phylogeny (Fig. 2), we suggest that a pericentric inversion or a centromere shift on the X chromosome occurred in the lineage that gave rise to *T. setosus* and *T. yonenagae*. The change in X chromosome morphology in the common ancestor of both species may be related to karyotype differentiation from other taxa and reproductive isolation. It has been suggested that chromosome rearrangements may affect chromatin structure (*Johnson & Lachance, 2012*) and, consequently, play a role in hybrid incompatibility. The change in gene expression after chromosome rearrangements was also suggested to contribute to the speciation process (*Potter et al., 2017*).

Our specimens and all the other species of the genus analyzed after CBG-banding, (*T. albispinus minor, T. iheringi, T. paratus,* and *T. yonenagae*) exhibited faint heterochromatic blocks on centromeric regions, mainly located on the smallest autosomes and the sex chromosomes (*Yonenaga-Yassuda et al., 1985*; *Leal-Mesquita et al., 1992*; *Fagundes, Camacho & Yonenaga-Yassuda, 2004*; *Lazar et al., 2017*). Interestingly, both cytotypes of *T. s. setosus* differed in heterochromatin distribution (Fig. 4), which may be involved in the chromosome evolution of this taxon.

*T. moojeni* and *T. s. setosus* had a large interstitial secondary constriction on the long arm of pair 6, which bears the NORs (Fig. 6). A chromosome pair with a large secondary constriction bearing the single NOR is a marker of echimyids, as already reported for *T. iheringi* and other Echimyidae genera (*Fagundes, Camacho & Yonenaga-Yassuda, 2004*; *Silva et al., 2012*; *Araújo et al., 2014*). The comparison of the GTG-banded karyotypes suggests that the NOR-bearing chromosome is the same in our specimens and in *T. albispinus, T. iheringi,* and *T. yonenagae* (*Leal-Mesquita et al., 1992*; *Fagundes, Camacho & Yonenaga-Yassuda, 2004*) and is probably conserved in the genus.

Hybridization with the telomeric probe showed signals only at the extremities of all chromosomes (Figs. 6C, 6F, and 6I). This pattern of hybridization was similar to that described for *T. iheringi* (*Fagundes, Camacho & Yonenaga-Yassuda, 2004*). *Bolzán (2017)* suggested that the absence of interstitial telomeric sequences indicates the evolutionary status of the chromosomes of a species. Accordingly, species without or with only a few interstitial telomeric sequences would have more conserved chromosomes, as seems to be the case of *Trinomys*.

## CONCLUSIONS

In summary, based on the available data, it is clear that the *Trinomys* species present conserved karyotypes with small variation in diploid numbers ($2n = 54$ to $2n = 61–65$) and mostly composed of biarmed autosomes. The X chromosomes are usually large submetacentrics and all the species analyzed presented one marker chromosome pair with a secondary constriction corresponding to the NOR, which is also typical for the other echimyid genera. The great conservation extends to the GTG- and CBG-banding patterns in the few species which had these patterns described. As previously proposed by *Leal-Mesquita et al. (1992)*, pericentric inversions, centromere repositioning, and other

minor rearrangements seem to be responsible for the chromosome evolution in this genus. Further analyses, including a robust phylogenetic hypothesis, cytogenetic studies with high resolution banding patterns and molecular data of a larger array of *Trinomys* species, are needed to improve our understanding of the chromosome evolution and genome organization of this genus. It should be stressed that *Trinomys* species, especially those from Minas Gerais, need more thorough morphological and molecular analyses, as their cytogenetic information alone is insufficient for taxonomic identification. In fact, several different species present very similar karyotypes (*Lazar et al., 2017*).

## ACKNOWLEDGEMENTS

The authors are indebted to Adriano Pereira Paglia, Beatriz Dias Amaro, Flávia Nunes Vieira, and Karla Leal who provided us with *Trinomys* material.

### Funding

This work was supported by a grant from Fundação de Amparo à Pesquisa do Estado de Minas Gerais (FAPEMIG) to Marta Svartman (Process APQ-02353-14). Naiara Pereira Araújo and Cayo Augusto Rocha Dias are supported with, respectively, postdoctoral and doctoral fellowships from Coordenação de Aperfeiçoamento de Pessoal de Nível Superior (CAPES). Marta Svartman is a recipient of a senior postdoctoral fellowship from CAPES. The funders had no role in study design, data collection and analysis, decision to publish, or preparation of the manuscript.

### Grant Disclosures

The following grant information was disclosed by the authors:
Fundação de Amparo à Pesquisa do Estado de Minas Gerais (FAPEMIG): APQ-02353-14.
Coordenação de Aperfeiçoamento de Pessoal de Nível Superior (CAPES).

### Competing Interests

The authors declare there are no competing interests.

### Author Contributions

- Naiara Pereira Araújo conceived and designed the experiments, performed the experiments, analyzed the data, prepared figures and/or tables, authored or reviewed drafts of the paper, approved the final draft, wrote the paper.
- Cayo Augusto Rocha Dias analyzed the data, prepared figures and/or tables, approved the final draft, wrote the paper.
- Rodolfo Stumpp analyzed the data, prepared figures and/or tables, approved the final draft.
- Marta Svartman conceived and designed the experiments, contributed reagents/materials/analysis tools, authored or reviewed drafts of the paper, approved the final draft, acquired funding, wrote the paper.

## Animal Ethics

The following information was supplied relating to ethical approvals (i.e., approving body and any reference numbers):

The conducted research was approved by the Ethics Committee in Animal Experimentation (CEUA) of Universidade Federal de Minas Gerais (approval number: 211/2013).

## Field Study Permissions

The following information was supplied relating to field study approvals (i.e., approving body and any reference numbers):

The permits were provided by the Instituto Chico Mendes de Conservação da Biodiversidade (ICMBio; permit number 22279-1 to Beatriz Dias Amaro) and the Instituto Brasileiro do Meio Ambiente e dos Recursos Naturais Renováveis (SISBIO-IBAMA; permit numbers 12989-2 and 36574-1, conceded to Adriano Pereira Paglia and Fabíola Keesen Ferreira, respectively).

## Data Availability

GenBank: MG214347, MG214348, and MG214349.

## Supplemental Information

Supplemental information for this article can be found online at http://dx.doi.org/10.7717/peerj.5316#supplemental-information.

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
