# Peer review of "Cytogenetic analyses in Trinomys (Echimyidae, Rodentia), with description of new karyotypes"

_PeerJ, doi:10.7717/peerj.5316_

## Round 0.1 · original submission · Major Revisions

Dear Naiara and Marta,

Both reviewers agree that this manuscript is a important contribution to Biodiversity knowledge of genus Trinomys (Echimyidae, Rodentia) but also agree that significant improvements should be performed.

Please, improve your manuscript following suggestions from both reviewers and submit a new manuscript that we will consider for publication in PeerJ.

Thanks!

Sincerely,
Marcial.

Reviewer 1 ·

Basic reporting

Major comments
This is a simple and short communication article, which provides a description of a new karyotype for the genus Trinomys. Molecular analyses were performed to check taxonomically the specimens. The manuscript highlights that Neotropical biodiversity is still poorly known. Methods described are important and with sufficient detail and information to be replicated, however morphological information was not investigated and it should be done to give more support to the discussion.

The title and abstract highlight a cytogenetic study, but the item Material and Methods starts inverted - in opposition to the main idea of the manuscript. The authors should consider the possibility of describing firstly the cytogenetic information and secondly the molecular information. It would be easier to describe karyotypes and then correlate them to the names - the trees should carry diploid numbers and the species names.
A map should be included in the manuscript with all the localities from where the sequences are and the distribution of the species as well.
The authors should remove Table 3 from the manuscript because all the information is in the Figure 4. In this figure, by the way, they should also check the G-banding pattern comparison of the triplet chromosomes 4; 7, 5, 5; 9; 12, 7, 7; 13; 15.

Experimental design

The cytogenetic and molecular methods used are adequate. To really improve the manuscript, as mentioned in the previous item, morphological studies should be included to improve the article. Additionally, FISH using specific probes could clarify what G-bands do not reveal (synteny).

Validity of the findings

According to the discussion (lines 179-181), the authors mentioned that the differences in the morphology of pairs 26 and/or 27 may be real or may reflect different chromosome condensation. This is an important point because if this is not real, the FN is the same for all individuals and some points are supposed to be reviewed.

Regarding the sex chromosome variation, other rodent families/subfamilies/tribes show X and Y polymorphisms. The discussion should be improved, for example, evaluating the several mechanisms evolved in the evolution of these chromosomes - addition/deletion of constitutive heterochromatin, centromere repositioning, for instance, and the cytogenetic alterations it could (or not) generate. Would it lead to a speciation process?

Additional comments

Minor comments
The authors should mention that Minas Gerais represents one state of Brazil and the number of metaphases counted per individual as well.
Figure 2 is indicated with letter A and B. Why do not use the same criteria in the Figure 3?
It is worth to insert the collecting localities in the legend of the figures.
The legend of Figure 3 should be corrected: Karyotypes of Trinomys setosus: a female with 2n=56, FN=106 (on the left) from Serro, Minas Gerais State, and of a male with 2n=56, FN=108 (on the right) from (locality) after GTG- (on the top) and CBG-banding (below). Scale bar = 10 μm.
The individual MCN-M 3296 (Table 2) is a male or a female?

Reviewer 2 ·

Basic reporting

The manuscript is an important contribution to the study of the genus Trinomys. It is well written and with a complete bibliographic research. I have a few suggestions and considerations.

In-text citations need to be checked. Looking at the authors instructions it is stated that: "for three or fewer authors, list all author names (eg Smith, Jones & Johnson, 2004)." For example the citations in the introduction: Emmons et al. 2015 and Dunnum et al. 2001 have 3 authors. And when there are two authors, the "&" should be used instead of "Lara and Patton 2000", "Carvalho and Salles 2004", for example.

Tables and Figures:

In table two could have another column with the Genbank accession number.

In relation to the figures, I understand that the manuscript already has many but I think it would contribute a lot if it had a map with the collection localities. This would be interesting even to see where are the localities of T. setosus and compare with the distribution of the two subspecies.

It would be interesting to have in Figure 1 the corresponding karyotype for the specimens included in the study. Otherwise, we have to go into the supplementary files to see which karyotype corresponds to which sequence. And and the phylogenetic tree (although based on only 401 bp) seems to show a structure in T. setosus. Do karyotypes confirm this? or not? This part could be more discussed as well.

The figure 3 legend is confusing; first replace “Metaphases” with “Karyotypes”. The title legend says: “Metaphases of a female Trinomys setosus (2n=56, FN=106)” but the figure shows both karyotypes (2n=56, FN=106, FN=108) of the female and the male.
In Supplementary Figure S3, why are T. setosus specimens in different colors?

Experimental design

Regarding molecular sequencing I have some issues about sequencing only 401 bp. But I understand that the tool was used only for specimen identification, and the results were congruent with previous studies.
Three of the five specimens were sequenced in the study. But as I could see in Supp. Files, it seems that the other two (T. setosus KX655539.1 (UFMG 6024) and T. moojeni KX650080.1 (MCN-M2816)) had previously been sequenced and deposited in GenBank. I suggest that this information should be included in the Material and Methods section.

Validity of the findings

In the results section, lines 139-140 says: “…and composed of 26 pairs of biarmed (pairs 1-26) and one pair of acrocentric (pair 27) autosomes”
but in line 143: “… that differed from those described above in the morphology of one of the smallest autosomes (pair 26) that was biarmed instead of acrocentric in these three animals” I guess it is pair number 27 instead of 26?
But, again, in lines 148-150 we see: “… and to verify that the complements of T. setosus with FN=106 and FN=108 (Fig. 4, Table 3) differed in relation to pair 26, which was acrocentric or biarmed, in the animals with FN=106 and FN=108, respectively.”

Line 163: Replace (Corrêa et al. 2005; Fig. 2) with (Corrêa et al. 2005; Fig. 1).

Pessoa et al. (2015) report that there is an unpublished karyotype for T. setosus: "The karyotype of T. s. setosus from Almenara, Minas Gerais state, is 2n = 56 and FN = 108 (unpublished data)." Although it does not have the published karyotype, the information is present in the last revision of the genus. It is worth to include this information in the discussion, even because this karyotype was related to T. setosus setosus with a related locality, as the karyotype 2n = 56 and FN = 104 was related to T. s. elegans (Pessôa et al. 2015). I think this part of the discussion could be a bit more developed, taking into account the subspecies, the karyotypes localities and their 2n and NF.

I don’t understand line 191 – The constitutive heterochromatin evidenced in our specimens was mainly located on the smallest chromosomes and the sex chromosome pairs in the two species” – as we can see in Figures 2 and 3 the first pairs are marked with constitutive heterochromatin (also written in lines 151- 153).

---

## Round 0.2 · Minor Revisions

Dear Naiara and Marta,
Both reviewers and I agree your manuscript has much improved and is almost ready to be accepted and published in PeerJ.
Could you please consider the several minor points raised by reviewer #1?
Thanks!
Sincerely,
Marcial.

Reviewer 1 ·

Basic reporting

The authors improved the manuscript, however I still have some considerations to be made.
Figure 1 - Coordinates and a kilometer bar should be inserted in the maps.

Figure 2 - The collection sites of the individuals could be included in this figure - at least the five animals of this study.

Concerning the lines 145-146 -- "Morphological characters analyzed corroborated the phylogeny and allowed assigning the T. setosus specimens to T. setosus" -- and lines 196-198 -- "Although our phylogenetic analyzes do not distinguish between T. s. setosus and T. s. elegans (Fig. 2, Supplemental Fig. S1 and S2), the karyotype and the morphological analysis seem important to recognize these subspecific taxa. ": The morphological analysis indicates T. setosus as a single entity? Cytogenetically, do the authors considered the karyotype as a diagnosable character for species identification (even considering the variation in the pair 26 and the similarity with the diploid and fundamental numbers of T. moojeni? Is that similar to T. gratiosus which also has 2n = 56 and FN = 108? Phylogenetically (Figure 2), the clade referred to T. setosus is composed of several relatively well-supported subclades, one of them (support 1) includes the animals from Morro do Pilar and Serro, with 2n = 56 and FN = 106 and FN = 108, due to variation in the pair 26 of autosomes; other clade (support 0.98) recovered the specimen from São Gonçalo do Rio Abaixo, which has 2n = 56 and FN = 108, besides the sister clade, for which th 2n is not shown. What is the distance between/among these localities, vegetation, topology, altitude, etc.?

Experimental design

Have the authors checked the presence of stop codon sequences? This detail can be crucial for alignment and analysis.

Validity of the findings

Figure 4 - It is really difficult to see that the pairs 26 in the Figures 4B and 4D are morphologically different. Are the authors absolutely sure that this happens?

Reviewer 2 ·

Basic reporting

All previous suggestions were accepted and improved

Experimental design

All previous suggestions were accepted and improved

Validity of the findings

All previous suggestions were accepted and improved

Additional comments

I would like to congratulate the authors. They accepted all previous suggestions and in my opinion the manuscript has been improved and should be accepted for publication. I have no further suggestions.

---

## Round 0.3 · accepted · Accept

Dear Naiara and Marta,

Congratulations!

Marcial.

#